The impact of COVID-19 pandemic on the level of depression among health care workers: cross-sectional study

http://orcid.org/0000-0002-9471-2767 Al Mutair Abbas 1 2 3
http://orcid.org/0000-0003-4181-6279 Al Mutairi Alya 4 amutairi@taibahu.edu.sa
Ambani Zainab 5
Shamsan Abbas 6
AlMahmoud Sana 7
http://orcid.org/0000-0003-4552-4513 Alhumaid Saad 8
1 Research Center, Almoosa Specialist Hospital , Alhasa , Saudi Arabia
2 College of Nursing, Princess Nourah Bint Abdulrahman University , Riyadh , Saudi Arabia
3 School of Nursing, University of Wollongong , Australia
4 Department of Mathematics, Faculty of Science, Taibah University , Medina , Saudi Arabia
5 King Saud Ben Abdulaziz University for Health Sciences , Al Ahsa , Saudi Arabia
6 Dr. Sulaiman Al Habib Medical Group , Riyadh , Saudi Arabia
7 Imam Abdurrahman Bin Faisal University , Riyadh , Saudi Arabia
8 Administration of Pharmaceutical Care, Al-Ahsa Health Cluster, Ministry of Health , Al-Ahsa , Saudi Arabia
Palazón-Bru Antonio
Electronic publication date: 2021 May 17
Publication date: 2021
Volume: 9
Electronic Location ID: e11469
Received 2021 Feb 17; Accepted 2021 Apr 26
Copyright: © 2021 Al Mutair et al.
Copyright year: 2021
Copyright holder: Al Mutair et al.
License: This is an open access article distributed under the terms of the Creative Commons Attribution License, which permits unrestricted use, distribution, reproduction and adaptation in any medium and for any purpose provided that it is properly attributed. For attribution, the original author(s), title, publication source (PeerJ) and either DOI or URL of the article must be cited.
License URL: https://creativecommons.org/licenses/by/4.0/

Keywords: COVID-19, Depression, Mental health, Sleeping disorder, Health care workers, Health care facilities, Saudi Arabia

Funding: The authors received no funding for this work.

==============================
Background

The outbreak of the novel Corona Virus Infectious Disease 2019 (COVID-19) has spread rapidly to many countries leading to thousands of deaths globally. The burden of this pandemic has affected the physical and mental health of the frontline health care workers (HCWs) who are exposed to high risk of infection and psychological stressors.

Aims

The aim is to measure the level of depression among healthcare workers in Saudi Arabia during COVID-19 pandemic to establish interventional strategies.

Method

A descriptive cross-sectional study was used to conduct the current study. The data of this study was recruited between 15 June and 15 July 2020 from healthcare providers who work in both public and private healthcare sectors in Riyadh and Eastern province in Saudi Arabia utilizing a self-administered questionnaire. The study was approved by the Institutional Review Board at Dr. Sulaiman Al Habib Medical Group (IRB Log No. RC20.06.88-2). Data were collected by using The Zung Self-Rating Depression Scale SDS. A total of 900 healthcare providers working in the healthcare setting during COVID-19 pandemic were invited to participate in the study. A total of 650 healthcare providers participated in the study by completing and submitting the survey.

Results

Almost 30% suffered from depression which can be divided into three categories; mild depression (26.2%), moderate/major (2.5%) and severe/extreme (0.8%). The finding shows that the level of depression among respondents at the age range of 31–40 years old was significantly higher than the level of depression among respondents with the age above 50 years old. Non-Saudi healthcare workers experienced more depression than Saudi workers. It also shows how nurses suffered from depression compared to their physician colleagues. Those who did not suffer from sleeping disorder perceived more depression as compared to those who are having sleeping disorder.

Conclusion

It is recommended that health care facilities should implement strategies to reduce the prevalence of mental health problems among healthcare providers and eventually it will improve their performance in provision of safe and high-quality care for patients.

Background

Mental health is one of the important issues that receives attention from the World Health Organization (WHO) as it was included in the Sustainable Developmental Goals for 2030. Over the last decade, mental health problems increased by 13% worldwide (World Health Organization, 2021). The most common mental health conditions are depression and anxiety which cost around US $ 1 trillion every year from global economy (World Health Organization, 2020). Mental health problems have a serious impact on all aspects of life such as interacting with others, family members, co-workers and even with patients. Despite that, the global median expenditure is less than 2% of the governmental health expenditure. The spread of COVID-19 pandemic caused huge stress on general population, however, the case was worse for healthcare workers due to their close interaction with COVID-19 patients (World Health Organization, 2020).

Frontline healthcare workers (HCWs) are dealing with suspected and positive cases of COVID-19 patients on a daily basis which puts them at high risk of infection and death (Chirico & Magnavita, 2020). Such exposure has a substantial impact on their mental health (Yörük & Güler, 2020; Bohlken et al., 2020; Ran et al., 2020). One of the most common mental health problems is depression. Depression can cause significant physical, emotional and behavioral problems (González-Sanguino et al., 2020; Zhou et al., 2020; Zhou et al., 2020). More than two thirds of healthcare workers serving in Turkey experienced high level of depression (Şahin et al., 2020). Being a female, a nurse, and having a history of psychiatric illness were identified as risk factors for mental health problems (Şahin et al., 2020). Many mental conditions can be treated or minimized by adopting some changes in the workplace, at home or in the community. However, evidence-based decisions need to be taken prior to implementing any strategy. Therefore, studies in this field are needed in order to examine the nature of the mental issues and how to reduce it. Since the spread of COVID-19 pandemic, researchers have conducted numerous studies to assess the impact of the COVID-19 on the physical and mental wellbeing of individuals. The target population varies according to the purpose of the study. Some studies examined the impact of the disease on COVID-19 patients (Bo et al., 2020); some focused on general populations (Fullana et al., 2020; Gao et al., 2020; Zhang et al., 2020), and some studies were specific to frontline healthcare workers (Chen et al., 2020; Du et al., 2020; Lai et al., 2020).

The psychological burden of health care workers had received attention in research publications. Numerous studies reported consistent results of health workers’ stress, anxiety, and depressive symptoms due to COVID-19 (Shreffler, Petrey & Huecker, 2020). Healthcare workers experience more psychological symptoms than non-healthcare workers. These symptoms include insomnia, anxiety, depression and obsessive-compulsive (Zhang et al., 2020). Moreover, studies also showed that HCWs experience emotional exhaustion, burnout, stress, and suicide (Di Tella et al., 2020). A study conducted in the Second Xiangya Hospital in China where 13 medical staff were interviewed to assess the mental health care during COVID-19 pandemic. Results showed how nurses presented signs of psychological distress, yet they refused receiving any psychological help considering that not a problem. They focused on the need of more uninterrupted rest and enough protective equipment (Chen et al., 2020). When frontline HCWs in Wuhan itself where surveyed, findings showed that HCWs were under moderate to severe stress and many reported elevated anxiety and depression (Chen et al., 2020). Italian healthcare providers also demonstrated high anxiety and depression levels during COVID-19 pandemic (Di Tella et al., 2020). This was a cross-sectional study which surveyed healthcare providers including doctors and nurses utilizing quality of life and health-related Visual Analogue Scales, State-Trait Anxiety Inventory, Beck Depression Inventory and PTSD Checklist (Di Tella et al., 2020). Findings of the current Italian study found that female healthcare providers were found to have higher levels of depression symptoms (Di Tella et al., 2020).

According to previous published literature, working in hospitals in general was associated with depression symptoms among healthcare providers (Santarone, McKenney & Elkbuli, 2020). Depression symptoms were reported more during COVID-19 pandemic among frontline healthcare providers (Santarone, McKenney & Elkbuli, 2020). In addition, high burnout level among Saudi and non-Saudi healthcare providers, it is crucial to deal with the issue for public health in Saudi Arabia and UAE (Al-Omari et al., 2020). Healthcare providers are more vulnerable to depression due to high turnover, heavy workload and fear of infections due to contagious nature of disease such as COVID-19 (Al Mutair, Alhajji & Shamsan, 2021).

As this pandemic continues to spread, its physical, emotional and mental impact on Saudi population and globally continues contributing to increase depression and emotional level (Al Mutair et al., 2017). Working in healthcare facilities have high amounts of psychological distress in doctors, nurses, and other health-care providers (Huang et al., 2020). Results of previously conducted study in Saudi Arabia showed that healthcare workers had high level of depression and anxiety indicating greater psychological distress. In the study female healthcare workers had more depression than their male colleagues, physicians had lesser psychological distress compared to other healthcare providers and Non-Saudi had higher psychological distress than Saudi professionals (Huang et al., 2020). COVID-19 has spread all over the world affecting many individuals’, especially healthcare providers. Previous studies have shown that clinicians in China suffered mental health problems from COVID-19 pandemic (Lee et al., 2007). In one of the studies which was conducted in 34 Chinese hospital surveyed 1,257 healthcare workers found that frontline healthcare workers and female nurses had higher risk of developing mental health issues and needed psychological support or medical interventions. Reports have also revealed that healthcare providers suffered from mental health problems during Sever Acute Respiratory Syndrome SARS (McAlonan et al., 2007; Lai et al., 2020; Al Mutair & Ambani, 2020; Al Mutair et al., 2020; Chirico & Nucera, 2020; Chirico, Nucera & Magnavita, 2021; Pappa et al., 2020). Similarly, the prevalence of mental illnesses during Middle East Respiratory Syndrome (MERS) epidemic was high among healthcare providers in Saudi Arabia (Al-Omari et al., 2020). Previous studies showed high prevalence of COVID-19 cases in Saudi Arabia ranging from mild to severe characteristics (Alenazi et al., 2020; Al Mutair et al., 2020). Furthermore, during COVID-19 pandemic studies have also shown that anxiety among healthcare providers in Saudi Arabia was high due to looking after COVID-19 patients (Xiao et al., 2020). Due to the lack of social support and communication, maladaptive coping, and lack of training on how to deal with such pandemic were identified as risk factors for developing psychological problems such as depression, anxiety, insomnia and distress, especially to those healthcare workers who interact directly with suspected and confirmed COVID-19 patients (Chen et al., 2020). Individuals with pre-existing psychiatric disorders reported deterioration of their mental health conditions due to COVID-19 (Chirico & Magnavita, 2020). Level of social support for medical staff is a significant predictor for self-efficacy and sleep quality and negatively associated with anxiety and stress (Arafa et al., 2021). The ongoing COVID-19 pandemic along with the measures to control it can lead to mental issues such depression. While the cases of COVID-19 attending healthcare centers and hospitals has been rapidly increasing along with extended hours of work can place huge mental pressure on healthcare providers. Given the cultural variations in responding to pandemics, it is highly important to investigate how healthcare providers in Saudi Arabia react emotionally and mental to COVID-19. This may help the government to initiate supportive programs to decrease and control healthcare providers depression and anxiety which may contribute to improve their psychological well-being. This study intends to investigate the level of depression among healthcare providers in Saudi Arabia

Methods

Study design

A descriptive cross-sectional non-interventional design was used for the purpose of this study. Anonymous self-administered, paper and pencil survey was used to collect data from the targeted setting. The researchers obtained the ethical approval to conduct the study from the Institutional Review Board at Dr. Sulaiman Al Habib Medical Group (IRB Log No. RC20.06.88-2). Participation in the study was voluntary and participants were ensured that information gathered for the study would be kept confidential and will be used for the study purposes only and consent letter was requested for this study as well. Level of depression is the dependent variable in the current study which will be measured by the Zung Self-Rating Depression Scale SDS. The soci-demographic profile of the participants is the independent variable. The initial sample size was estimated using G*Power3 and based on multiple linear regression using the independent two-tailed t-test, confidence level of 95%, margin rate of error at 5% and power of 80.0%, medium effect size of 0.30 (determined based on the review of current literature), and a 10% increase to address the non-response rate, the minimum required sample size for this study was 353 subjects. A total of 900 surveys were distributed among healthcare providers, 650 returned back the completed surveys giving a response rate of 72%.

Saudi and non-Saudi healthcare providers were eligible to be a participant in this study if all of the following criteria are met: (1) is 18 years of age or older (2) works in inpatient or outpatient healthcare setting (3) is responsible for providing direct bedside patient care, and (4) has spent at least 3 months in current healthcare unit. Healthcare providers who work in private and public healthcare sectors in Riyadh, Saudi Arabia were invited to participate in the study by completing and returning back the questionnaire. Data collection continued for 2 weeks between 2 to 15 April, 2020.

Data collection instrument

The Zung Self-Rating Depression Scale SDS was used to collect data from the participants. It consists of 20 items which has been widely used as a screening tool for depression in multiple populations. The SDS statements are framed in a positive and negative pattern with 4-Likert point scale ranging from 1 a little of the time, 2 some of the time, 3 good part of the time and 4 most of the time. The scale score may range from 20 to 80, scores from 20 to 49 indicating no depression, 50 to 69 indicating depression and scores from 70 to 80 indicting severe depression (Dunn & Sacco, 1989). The scale has shown good validity and acceptable reliability for clinical and research purposes with alpha value 0.84 (Dunn & Sacco, 1989). Additional part to the questionnaire about the socio-demographic characteristics of the respondents was added in the current study. This included age, gender, marital status, economic status, nationality, profession, working area, years of working experience and type of healthcare facility.

Data analysis

Frequencies and percentages for nominal/ordinal level variables was employed. Chi-square analysis has been conducted to determine the association between demographic profile and depression. Unpaired Independent t-test and one-way ANOVA analysis to compare healthcare providers’ level of depression and predict the factors of depression among the study populations. P-value of ≤ 0.05 were accepted as the significance level for all inferential statistical tests.

Results

Demographic profile analysis

Demographic profile has been analyzed and the findings are presented in the Table 1. Several demographic profiles have been chosen namely the type of health care facility, age, gender, nationality, health specialty, hospital department, years of experience, sleeping disorder before COVID-19 and mental disorder. In terms of type of health care facilities, 444 (68.7%) of participants work in private facilities while only 202 (31.3%) work in governmental hospitals. By looking at the age, about 177 (46.6%) of the respondents were at the age range 31–40 years old, followed by 123 (32.4%) were 20–30 years old, 63 (16.6%) were 41–50 years old and a few number of workers 17 (4.5%) were above 50 years old. The percentage of females (n = 475, 74%) was larger than the percentage of males (n = 167, 26%). Almost three quarters of the respondents 450 (70.4%) were non-Saudis while 189 (26%) were Saudis. By studying at the health specialty, 323 (50.2%) of the respondents were nurses, 114 (17.7%) were physicians, and 207 (32.1%) were from other specialties. Approximately 243 (38.3%) of the respondents work in Intensive Care Unit (ICU), followed by 206 (32.4%) work in other departments, 134 (21.1%) work in wards and 52 (8.2%) work in Emergency department. By studying at the year of experience, 280 (44.3%) have 1 to 5 years of work experience, followed by 177 (28%) with 6 to 10 years of work experience and 175 (27.7%) with over 11 years of work experience. Respondents were asked about sleeping disorder before COVID-19, about 532 (86.9%) did not have sleeping disorder while 80 (13.1%) had sleeping disorder before COVID-19. Lastly, in terms of mental disorder, about 592 (96.7%) did not have mental disorder, while 20 (3.3%) had mental disorder.

Table 1 Demographic profile of the respondents.

Demographic Profile	n	%	
Type of health care facility			
Government	202	31.3	
Private	444	68.7	
Age			
20–30 Years Old	123	32.4	
31–40 Years Old	177	46.6	
41–50 Years Old	63	16.6	
Above 50 Years Old	17	4.5	
Gender			
Male	167	26.0	
Female	475	74.0	
Nationality			
Saudi	189	29.6	
Non-Saudi	450	70.4	
Health specialty			
Physicians	114	17.7	
Nurses	323	50.2	
Others	207	32.1	
Hospital Department			
ER	52	8.2	
Wards	134	21.1	
ICU	243	38.3	
Others	206	32.4	
Years of experience			
1–5 Years	280	44.3	
6–10 Years	177	28.0	
11 Years and Above	175	27.7	
Sleeping disorder before COVID-19			
Yes	80	13.1	
No	532	86.9	
Mental Disorder			
Yes	20	3.3	
No	592	96.7	

Level of depression

The Level of depression has been divided into four categories as shown in Table 2, which they are normal, mild depression, moderate/major and severe/extreme. Almost over three quarters of respondents did not suffer from depression which comprised 459 (70.6%). Furthermore, the findings revealed less than 30% suffer from depression which falls into three categories; mild depression 170 (26.2%), moderate/major 16 (2.5%) and severe/extreme 5 (0.8%) as shown in Fig. 1.

Table 2 Level of depression.

Level	Frequency	Percent	
Normal	459	70.6	
Mild depression	170	26.2	
Moderate/Major	16	2.5	
Severe/Extreme	5	0.8	
Total	650	100.0	

Figure 1 Level of depression among study participants.

The association between demographic profile and depression

Chi-square analysis has been conducted to determine the association between demographic profile and depression as shown in Table 3. Several demographic profiles have been selected which are health care facilities, age, gender, nationality, health specialty, work area, years of experience, sleeping disorder before COVID-19 and mental disorder. The findings have presented in Table 4 further revealed age (X2 = 12.89, df = 6, p < 0.05), nationality (X2 = 8.321, df = 2, p < 0.05), health specialty (X2 = 9.54, df = 4, p < 0.05) and sleeping disorder before COVID-19 (X2 = 12.127, df = 2, p < 0.05) have significant association with depression. However, healthcare facility (X2 = 2.377, df = 2, p > 0.05), gender (X2 = 3.683, df = 2, p > 0.05), hospital department (work area) (X2 = 10.216, df = 6, p > 0.05), years of experience (X2 = 7.134, df = 4, p > 0.05) and mental disorder (X2 = 5.919, df = 2, p > 0.05) insignificantly influenced depression. The findings showed that among respondents at the age range of 31-40 years old, the level of depression was (46.6 ± 39.59) significantly higher than respondents with the age above 50 years old (4.5 ± 32.37). In terms of nationality, non-Saudi (70.4 ± 40.68) suffered from depression compared to Saudi (29.6 ± 40.08). Other healthcare providers’ specialties (32.1 ± 41.70) were more depressed than physicians (17.7 ± 39.70). Those who did not suffer from sleeping disorder (86.9 ± 39.93) perceived depression compared to those who are having sleeping disorder (13.1 ± 41.20).

Table 3 The Association between demographic profile and depression.

Variables	Normal	Mild	Moderate	Severe	Value	df	Sig.	
N	%	N	%	n	%	n	%	
Type of health care facility									2.377	2	0.305	
Government	152	75.2	40	19.8	9	4.5	1	0.5	
Private	306	68.9	127	28.6	7	1.6	4	0.9	
Age									12.892	6	0.045	
20–30	94	76.4	25	20.3	4	3.3	4	2.3	
31–40	133	75.1	33	18.6	7	4.0	4	2.3	
41–50	53	84.1	9	14.3	1	1.6			
Above 50 years	15	88.2	2	11.8					
Gender									3.683	2	0.159	
Male	129	77.2	35	21.0	1	0.6	2	1.2	
Female	329	69.3	128	26.9	15	3.2	3	0.6	
Nationality									8.321	2	0.016	
Saudi	130	68.8	50	26.5	8	4.2	1	0.5	
Non Saudi	325	72.2	113	25.1	8	1.8	4	0.9	
Health specialty									9.543	4	0.049	
Physicians	83	72.8	21	18.4	8	7.0	2	1.8	
Nurses	237	73.4	78	24.1	5	1.5	3	0.9	
Others	137	66.2	67	32.4	3	1.4			
Hospital Department									10.216	6	0.116	
ER Department	35	67.3	13	25.0	3	5.8	1	1.9	
Ward	98	73.1	32	23.9	4	3.0			
ICU Department	173	71.2	58	23.9	8	3.3	4	1.6	
Others	148	71.8	57	27.7	1	0.5			
Years of experience									7.134	4	0.129	
1-5Years	206	73.6	68	24.3	5	1.8	1	0.4	
6-10Years	111	62.7	56	31.6	7	4.0	3	1.7	
11 Years and Above	136	77.7	35	20.0	3	1.7	1	0.6	
Sleeping disorder before COVID-19									12.127	2	0.002	
Yes	55	68.8	19	23.8	5	6.3	1	1.3	
No	394	74.1	123	23.1	11	2.1	4	0.8	
Mental Disorder									5.919	2	0.052	
Yes	10	50.0	8	40.0	2	10.0			
No	438	74.0	135	22.8	14	2.4	5	0.8	

Table 4 Descriptive analysis (frequency, percentage, mean and SD).

Demographic profile	N	%	Mean	SD	
Type of health care facility					
Government	202	31.3	37.99	13.97	
Private	444	68.7	41.74	11.51	
Age*					
20–30 Years Old	123	32.4	38.12	12.79	
31–40 Years Old	177	46.6	39.59	13.76	
41–50 Years Old	63	16.6	36.98	11.07	
Above 50 Years Old	17	4.5	32.37	11.38	
Gender					
Male	167	26.0	38.01	12.91	
Female	475	74.0	41.37	12.18	
Nationality*					
Saudi	189	29.6	40.08	13.79	
Non Saudi	450	70.4	40.68	11.89	
Health specialty*					
Physicians	114	17.7	39.70	14.30	
Nurses	323	50.2	40.09	12.10	
Others	207	32.1	41.70	11.85	
Hospital Department					
ER Department	52	8.2	41.84	13.50	
Ward	134	21.1	40.06	11.58	
ICU Department	243	38.3	38.83	14.23	
Others	206	32.4	42.26	10.12	
Years of experience					
1–5 Years	280	44.3	40.93	11.82	
6–10 Years	177	28.0	42.25	13.03	
11 Years and Above	175	27.7	37.53	12.46	
Sleeping disorder before COVID-19*					
Yes	80	13.1	41.20	15.18	
No	532	86.9	39.93	12.04	
Mental disorder					
Yes	20	3.3	43.91	15.27	
No	592	96.7	40.07	12.35	
Note:

* Significance at alpha 0.05.

Discussion

Depression and anxiety are common and cited problems that generally encounter individuals and more specifically frontline healthcare workers (Al-Omari et al., 2020; Lai et al., 2020). Our study aimed to measure the level of depression among healthcare workers in Saudi Arabia during COVID-19 pandemic and to measure the level of depression based on demographic profile. Several limitations should be taken into account while interpreting the results of the current study. A cross-sectional design was used to recruit healthcare providers from Riyadh capital city in Saudi Arabia, so caution must be practiced while generalizing its results to all healthcare providers in the county. Additionally, current findings rely heavily on subjective perception of healthcare providers rather than objective methods or clinical diagnosis and therefore further studies on prevalence of depression among healthcare providers should be conducted subjectively during and after the pandemic.

In our study, the prevalence of depression among healthcare workers was 29.5% which is close to its percentage in Turkey (31.8%) (Lai et al., 2020). Since the spread of COVID-19 in Wuhan and Hubei in China, psychological distress was reported in 70% of healthcare workers (Zhou et al., 2020). In Iran, the prevalence of depression among HCWs was reported as 30% (Al Mutair & Ambani, 2020). The classifications of depression levels in this study is similar to other studies that found the higher percentage of those with depression symptoms had mild depression, followed by those with moderate depression, and the least percentage was the HCWs who suffer from severe depression (Lai et al., 2020). For instance, in other studies, healthcare providers from Saudi Arabia and Egypt reported high level of depression, anxiety and stress during COVID-19 pandemic (Arafa et al., 2021). Young female healthcare providers had scored higher on depression and stress scales especially those who were recruited from Egypt. Researchers have suggested psychological support interventions programs targeting healthcare providers.

It is expected that the percentage in China is higher than in other countries due to lack of information on how to deal with this new virus at the beginning of its onset. Upon the spread of the infection to other countries, some information about the virus and precautions became available which minimized the uncertainty. The fear of being infected by COVID-19 is the core cause of anxiety and other mental health problems. Some of the reported risk factors for that are spending long hours in hospitals with COVID-19 patients, suboptimal hand cleanliness, and lack of protective equipment (Chen et al., 2020; Alenazi et al., 2020). The findings show that the level of depression was significantly high for HCWs whom age is between 31 to 40 years respondents compared to those who are above 50 years old. This could be related to social responsibilities for this age group who are usually married, have children or family members who are at old age and considered as the most vulnerable group. Nationality variable was not studied previously in similar studies. The results show a significantly higher level of depression among non-Saudi HCWs as compared to Saudi HCWs. In addition to the stress and anxiety associated with working in hospitals with COVID-19 patients, non-Saudi encounter more stressors such as not being able to visit their families, return to their home countries, the fear and worries of getting infected, facing death alone, or having a family member dying while they are away from home. These tragedies have happened in reality and they are leading to higher levels of anxiety and depression to non-Saudi HCWs. Consistent with other studies, nurses are experiencing more depression than other HCW (Chen et al., 2020). Although physicians are dealing with COVID-19 patients, nurses are closer to the patients in terms of interaction and spending longer periods. Besides the exposure to this virus, this could create more psychological stress and worries which would lead to depression. Our findings of the association between having previous sleep disorders and depression is slightly different than other studies. Our results show that those who did not suffer from sleeping disorder perceived more depression than those who are having sleeping disorder. In contrast, other studies found that having previous mental health or neurological conditions has been associated with increase of depressive and anxiety symptoms (Zhou et al., 2020; Şahin et al., 2020). Moreover (Şahin et al., 2020; Bo et al., 2020; Fullana et al., 2020) these studies had also found that individuals with pre-existing psychiatric problems reported more mental health problems during COVID-19 pandemic. In our study, there was an association between having previous mental health problems and depression where it is very close to significant association (p = 0.052). In general, the study findings suggest that COVID-19 pandemic has a negative impact on the mental health of HCWs as it exposes them at risk of developing depression and insomnia. Health care systems need to focus not only on caring for COVID-19 patients, but also on those who provide care for them in order to continue the care provision effectively and safely. Some of the suggested interventions are:Develop strategies in the workplace to optimize the mental health of HCWs such as creating places to exercise, mindfulness, meditation, and places for rest and sleep especially for those who work long hours.

Ensure availability of Personal Protective Equipment (PPE) to all HCWs. This will reduce the worries of being infected and will increase the sense of security and safety

Monitor strict adherence to all safety precautions while dealing with patients and colleagues at workplace.

Modify the HCWs’ work schedule to have shorter shifts to allow workers to rest and socialization with their families.

Allow HCWs to use technology and telehealth when there is no need for direct contact with patients to minimize the exposure to infection.

Encourage HCWs to seek help and consultation whenever they start feeling any anxiety, depression, insomnia or other mental health problems. This will reduce the complication of cases and help them receiving the appropriate treatment earlier.

These interventions may improve the mental health of the HCWs and prevent further complications if they were addressed early. The current study findings should be interpreted taking into full considerations its limitations. The study sample is composed represent private facilities more than government hospital (68.7% vs. 31.3% respectively), and the percentage of non-Saudis were higher than Saudis (70.4% vs. 29.6% respectively). This may limit the generalization of results.

Conclusion

Overall, level of depression has been divided into four levels. The findings revealed less than 30% suffer from depression which fall into three categories; mild depression 170 (26.2%), moderate/major 16 (2.5%) and severe/extreme 5 (0.8%). Several demographic profiles have been selected which are healthcare facilities, age, gender, nationality, health specialty, hospital department, years of experience, sleeping disorder before COVID-19 and mental disorder. The findings further revealed age, nationality, health specialty and sleeping disorder before COVID-19 have significant association with depression. However, healthcare facilities, gender, hospital department, years of experience and mental disorder insignificantly influenced depression. Those who did not suffer from sleeping disorder perceived more depression as compared to those who are having sleeping disorder. It is recommended that health care facilities develop and implement some strategies to reduce the prevalence of mental health problems especially depression and insomnia. The impact of these strategies may improve the mental health of HCWs and eventually improve their performance in provision of safe and high quality care for patients.

Supplemental Information

Supplemental Information 1 Raw measurements.

Click here for additional data file.

The authors thank the referee for constructive comments.

List of abbreviations

COVID-19 Coronavirus disease 201

SARS-CoV-2 Severe acute respiratory syndrome coronavirus 2

WHO World Health Organization

IRB Institutional Review Board

HCWs Health care works

SPSS Statistical Package for the Social Sciences

SD Standard deviation

IQR Interquartile range

SDS Self-Rating Depression Scale

Additional Information and Declarations

Competing Interests

Author Contributions

Ethics

Data Availability

Abbas Al Mutair and Abbas Shamsan are employed by the Dr. Sulaiman Al Habib Medical Group. The authors declare no conflict of interest in preparing this article.

Abbas Al Mutair conceived and designed the experiments, authored or reviewed drafts of the paper, and approved the final draft.

Alya Al Mutairi performed the experiments, analyzed the data, prepared figures and/or tables, and approved the final draft.

Zainab Ambani performed the experiments, prepared figures and/or tables, and approved the final draft.

Abbas Shamsan conceived and designed the experiments, performed the experiments, authored or reviewed drafts of the paper, and approved the final draft.

Sana AlMahmoud performed the experiments, prepared figures and/or tables, and approved the final draft.

Saad Alhumaid conceived and designed the experiments, performed the experiments, prepared figures and/or tables, and approved the final draft.

The following information was supplied relating to ethical approvals (i.e., approving body and any reference numbers):

The study was approved by the Institutional Review Board in The Research Center, Dr. Sulaiman Al Habib Medical Group (IRB Log No. RC20.06.88-2).

The following information was supplied regarding data availability:

The raw measurements are available in the Supplemental File.

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
