# Peer review of "The impact of COVID-19 pandemic on the level of depression among health care workers: cross-sectional study"

_PeerJ, doi:10.7717/peerj.11469_

## Round 0.1 · original submission · Major Revisions

I think your work has some scientific merit, after reading the manuscript and the reviewers' comments. However, there are major issues which you should address in a revised version of the text.

Reviewer 1 ·

Basic reporting

The English language should be improved to ensure that an international audience can clearly understand your text.

Some examples where the language could be improved include line 254 – the current phrasing makes comprehension difficult (the term "Insignificantly").

Experimental design

In methods the authors should give more details about independent variables (sleeping disorder, mental health problems), on how they code and measure these variables

Validity of the findings

Findings are correct. Please, add the limitations of your study

Additional comments

Please, revise my comments. Furthermore, I suggest to improve for clarity and grammar the presentation of the work.

·

Basic reporting

a. Throughout the manuscript, English used is ambiguous and not clear. Also the manuscript has many typing errors

• Eg: 650 returned back the completed surveys 118 giving a response rate of 72%
• Depression and anxiety are common and cited problems that generally encounter individuals and more 188 specifically frontline healthcare workers
• COVID-19: Coronavirus disease 201; HCWs: Health care works;
• This The authors declare no conflict of interest

b. Referencing in the literature is not as per the standards of the journal

a. According to [11], lack of social support and communication, maladaptive coping, and lack of training on 94 how to deal with such pandemic were identified as risk factors for developing psychological problems 95 such as depression, anxiety, insomnia and distress especially to those healthcare workers who interact 96 directly with suspected or confirmed COVID-19 patients [11]
b. According to the study by [16], show a more satisfactory fit to the data than the results produced by 90 the (confirmatory factor analysis)
c. Overall study design, methodology and results have been poorly presented
Raw data supplied

Experimental design

1.Title: The Impact of COVID-19 Pandemic on the Level of Depression among Health Care Workers: Cross-sectional Study
The title of the study doesn’t give information about the site of the study and is misleading as in a cross sectional study no cause effect relation can be established. So, COVID-19 can’t be causally attributed to depression in frontline workers but only associated with it.
2. Aim and research question
The aim of the study as given in the abstract is different from that in the methodology. Identification of interventional strategies for depression in front line workers is only seen in the abstract. No attempt was made to design these. Till date many studies have been carried out to assess the psychological impact of COVID-19 on health workers, what is the need of the current study?? This is not clearly evident from introduction.
3. Inclusion criteria
The authors didn’t specify in their inclusion criteria about the exposure of included frontline workers to COVID-19, neither was the duration of working hours, specialty included in their assessment.
No objective measures were used for the assessment of mental disorder and sleeping disorder, how was the assessment done??
4. Results and discussion
a. Methodology shows the number of respondents to be 650 but table-1 shows socio-demographic characteristics none of which account to 650. Same has been observed in the text also.
Eg: Almost three quarters of the 152 respondents 450 (70.4%) were non-Saudis while 189 (26%) were Saudis
b. Findings contrary to the previous studies- Our results show that those who did not suffer from sleeping disorder perceived more depression
c. Authors show new results in discussion section, the assessment of which wasn’t shown anywhere in the methodology/results section
Eg: In addition to the stress and anxiety associated with working at hospitals with COVID-19 patients, non-Saudi encounter more stressors such as not being able to visit their families or to return to their home countries, the fear and worries of getting infected, facing death alone, or having a family member die while they are away

Validity of the findings

Impact and novelty not assessed.
Inconclusive results portrayed as positive.
Eg In our study, there was an association between having previous mental health problems and depression, although it is not significant, it is very close to significant 222 association (p= 0.052)

Conclusions are just mere rewriting of the results

Additional comments

Overall English needs to be improved. Authors need to stick to their aim and present their results and discussion accordingly.

Reviewer 3 ·

Basic reporting

This is a significant cross-sectional study reporting the prevalence of depression in Saudi Arabia healthcare workers. Nonetheless I have a number of concerns that I think must be considered and responded to, in order to strengthen the manuscript for potential publication. Please see below:
Comment 1: Needs a thorough English check.
Comment 2: Abstract - The methods need to be more informative. Please include where and when this study was conducted.
Comment 3: Background - the authors have pointed to the conducted studies in China, please, provide the results of the related systematic review and meta-analysis studies.
Comment 4: Background - Please give some examples of related studies, in Saudi Arabia and their results. Also, specify the differences between them and your study.
Comment 5: Background – Some sentences are hard to understand, please clarify:
• Working in hospitals in general was associated with clinically significant depressive symptoms whether the workplace is in COVID-19 associated department or other departments.
• According to the study by [16], show a more satisfactory fit to the data than the results produced by the (confirmatory factor analysis) CFA of the original version indicating that the newly constructed version can allow for a more accurate estimation of burnout levels.

Experimental design

Comment 6: Aim of the study – I think it is better to add a paragraph at the end of the Background section to describe the purpose and importance of the research.
Comment 7: Please insert a section titled “Methods” after the Background section.
Comment 8: Study design – Please specify where and when the study was conducted.

Validity of the findings

Comment 9: Needs a limitations section to address possible weaknesses before the Discussion section.
Comment 10: Discussion – Also, as suggested in Comment 3, compare the results with the other studies in Saudi Arabia. Also, it could be interesting to compare with the studies in other Arabian countries.

Annotated reviews are not available for download in order to protect the identity of reviewers who chose to remain anonymous.

---

## Round 0.2 · Minor Revisions

Still pending some minor modifications suggested by the reviewers.

Reviewer 1 ·

Basic reporting

1) There are still some minor typos (e.g. in introduction:
"Over the last decade, mental 56 health problems increased by 13% worldwide WHO, [1]. "
"The most common mental health conditions are 57 depression and anxiety which cost approximately US $ 1 trillion yearly from the global economy see 58 WHO, Mental Health, [2]."
"Therefore, studies in this filed are needed.."
2) Some sentenced in introduction need to be referenced.
"The spread of COVID-19 pandemic caused huge stress on general population, however, the case was worse for healthcare workers due to their close interaction with COVID-19 patients". "Frontline healthcare workers (HCWs) are dealing with suspected and positive cases of COVID-19 patients on daily basis which puts them at high risk of infection and death" (you can cite: Chirico F, Magnavita N. Covid-19 infection in Italy: An occupational injury. S Afr Med J. 2020 May 8;110(6):12944. Doi: 10.7196/SAMJ.2020.v110i6.14855."
"Such exposure has a substantial impact on their mental health" (you may cite: Chirico F, Nucera G, Magnavita N. Protecting the mental health of healthcare workers during the COVID-19 emergency. BJ Psych International. 2020. 1-6. Doi: 10.1192/bji.2020.39"

Experimental design

Aim of the research is clear. Methods are rigorous and are well described

Validity of the findings

All data sound statisticall robust

Reviewer 3 ·

Basic reporting

The authors have revised their manuscript to address most of my original points, which I believe has improved the manuscript. However, some of the changes have not satisfactorily solved the issue. Please see below:
Comment 1: Abstract - The authors inserted some information in this section but did not mention about the date and location conducted the study.
Comment 2: Background – There are a lot of studies in different countries around the world that evaluate the mental health problems of medical staff during the COVID-19 outbreak, I expect the authors to refer to some countries using the review studies, not just China.
Comment 3: Background – the authors mentioned to some studies in Saudi Arabia. I think it values to insert their results. Also, specify the differences between them and your study and explain the necessity of your study, please.
Comment 4: Aim of the study – I suggest delete this section.

Experimental design

No comment.

Validity of the findings

No comment.

Additional comments

No comment.

Annotated reviews are not available for download in order to protect the identity of reviewers who chose to remain anonymous.

---

## Round 0.3 · Minor Revisions

Still pending some minor modifications suggested by one of the reviewers.

Reviewer 1 ·

Basic reporting

English language could be improved
There are typographical mistakes (e.g. filed instead of field in the introduction),
"The soci-demographic profile" in methods
As suggested, background/context on mental health in HCWs should be improved (Cite:Chirico F, Nucera G. Tribute to healthcare operators threatened by COVID-19 pandemic. J Health Soc Sci. 2020;5(2):165-168. 10.19204/2020/trbt1. Chirico F, Nucera G, Magnavita N. Protecting the mental health of healthcare workers during the COVID-19 emergency. BJ Psych International. 2020. 1-6. Doi: 10.1192/bji.2020.39. Pappa S, Ntella V, Giannakas T, Giannakoulis VG, Papoutsi E, Katsaounou P. Prevalence of depression, anxiety, and insomnia among healthcare workers during the COVID-19 pandemic: A systematic review and meta-analysis. Brain Behav Immun. 2020 Aug;88:901-907. doi: 10.1016/j.bbi.2020.05.026. Epub 2020 May 8. Erratum in: Brain Behav Immun. 2021 Feb;92:247.)

Experimental design

The authors have addressed all my previous comments

Validity of the findings

The authors have addressed all my previous comments

Additional comments

Please, address some minor suggestions in introduction and revise the english language that may be improved.

Reviewer 3 ·

Basic reporting

No comment.

Experimental design

No comment.

Validity of the findings

No comment.

Additional comments

The authors have revised their manuscript in light of my previous comments and they have improved it to a level of satisfaction.

---

## Round 0.4 · accepted · Accept

The current version of your work has high standards to be published in PeerJ.

Congratulations!

Reviewer 1 ·

Basic reporting

The authors have addressed all my comments

Experimental design

The authors have addressed all my comments

Validity of the findings

The authors have addressed all my comments

Additional comments

Congratulations!
it's a good work!